# Virtual triage and outcomes of diabetic foot complications during Covid-19 pandemic: A retro-prospective, observational cohort study

**Ashu Rastogi**[1]*, **Priya Hiteshi**[1], **Anil Bhansali A.**[1], **Edward B. Jude**[2]*

**1** Department of Endocrinology, PGIMER, Chandigarh, India, **2** Tameside and Glossop Integrated Care NHS Foundation Trust, Tameside on Lyne, United Kingdom and Manchester Metropolitan University, Manchester, United Kingdom

* edward.jude@tgh.nhs.uk (EBJ); ashuendo@gmail.com (AR)

**Data Availability Statement:** All relevant data are within the manuscript and its Supporting information files.

## Abstract

### Aims

Limb and patient outcomes in people with diabetic foot complications including diabetic foot ulcer (DFU) provided virtual triage and personalized video consultations during COVID-19 pandemic are not known.

### Methods

Patients with foot complications attending the diabetic foot clinic prior to lockdown who sought teleconsultations during COVID-19 lockdown underwent virtual triage to include clinical history, visual inspection of feet, domiciliary wound care (community nurse assisted dressings) and offloading instructions. The subsequent ulcer, limb and mortality outcomes during the following 24 weeks of COVID-19 lockdown (April-September 2020, group 1) were assessed and compared with those who attended foot clinic during the same period in 2019 (April-September, group 2).

### Results

Group 1 included 561 participants with foot complications provided with teleconsultations, median age 57 (51 to 63) years and diabetes duration of 10 (5 to 16) years. Twelve patients with severe DFU were excluded and 549 patients [357 (65%) neuropathic foot, 104 (18.9%) ischemic foot and 88 (16%) chronic Charcot foot with deformities] were evaluated. There were 227 (41.3%) participants with active DFU at start of lockdown, 32 (5.8%) with new onset ulcer during lockdown (47.1%) and 290 patients without ulcers. Group 2 included 650 participants; active foot ulcer was present in 366 patients. Wound closed or reduced in area in 78.4% of participants of group 1 compared to 76.0% (p = 0.318) in group 2. Fourteen (5.4%) patients required amputations [3 major and 11 minor] in group 1 during the study period compared to 6.8% in group 2 (p = 0.191). Twenty-one (3.8%) and 28 (4.3%) patients died (p = 0.532) during 24 weeks of follow up in group 1 and 2, respectively.

**Funding:** The author(s) received no specific funding for this work.

**Competing interests:** The authors have declared that no competing interests exist.

## Conclusions

Targeted foot-care service through virtual triage and teleconsultations during COVID-19 pandemic for people with foot complications have similar ulcer and limb outcomes compared to face-to-face foot care delivery.

## Introduction

The coronavirus disease (COVID-19) was declared a global pandemic by WHO that has adversely affected the lives of millions of individuals. India is already a witness to an epidemic of non-communicable disease with more than 62 million individuals living with diabetes [1]. COVID-19 is known to be associated with poor outcomes in vulnerable population particularly amongst people with diabetes [2, 3]. COVID-19 has encumbered the overburdened healthcare workers and a drain on the resources from caring for people with diabetes and co-existing diabetic complications to the care of people afflicted with SARS-CoV-2. The COVID 19 pandemic is unprecedented and has affected routine care of people with diabetic foot complications [4]. Foot complications in diabetes encompass a spectrum ranging from the "at risk foot" to more severe diabetic foot disease including foot ulcer, diabetic foot infections, limb claudication, gangrene and Charcot neuroarthropathy. It has been shown that irrespective of the type of foot complications, these patients have higher mortality risk than those with without foot complications [5, 6].

Guidelines suggest annual foot examination in people with diabetes and more frequently in those with complications like foot ulcer or peripheral arterial disease (PAD) [7]. People with diabetic foot require periodic visits to hospital for appropriate foot care and or regular monitoring of existing foot complications especially diabetic foot infections (DFI) and foot ulcer (DFU) [8]. The global pandemic and subsequent lockdown to curb SARS-CoV-2 transmission has resulted in the suspension of non-emergent foot care services including preventative and conventional out-patient foot care that is essential for an early recognition and treatment of foot complications [9, 10]. Face to face consultations in most clinics have been reduced or are non-existent during COVID-19 pandemic perpetuated by lockdown [9]. However, in patients with diabetic foot it can be challenging because these patients require face-to-face clinic interaction with a foot care specialist for wound assessment, wound debridement and dressings, which is increasingly difficult during these times [11, 12].

The COVID-19 pandemic has necessitated a switch from traditional way of providing foot care services to unconventional methods of virtual service delivery including virtual triage, teleconsultations and remote patient monitoring. In a survey performed in India it was observed that most patients with diabetes prefer telecommunication through video calls during the COVID-19 pandemic to obtain physician consultation because of fear of acquiring SARS-CoV-2 infections and psychological adaptive difficulties [13, 14]. Telemedicine facility and remote methods for DFU assessment during pre-COVID times have been shown to be effective in preventing amputation in diabetic foot ulcers patients and are associated with similar ulcer and limb outcomes when compared to face-to-face consultation [15, 16]. However, data on DFU and limb outcomes for people with diabetes provided remote video consultation during the pandemic is sparse.

Recent guidance statements for diabetic foot disease do stress upon the utilization of teleconsultation for uncomplicated foot problems during the COVID-19 pandemic [4, 11, 17, 18]. In India, the government has implemented a telemedicine portal, e-Sanjeevani (www. eSanjeevani.in) that has made consultations for chronic ailments feasible for patients in remote

locations. The present prospective study was undertaken to determine the clinical characteristics and outcomes of limb and lives in people with diabetic foot complications who underwent virtual triage and supervised teleconsultations for foot complications during the COVID-19 pandemic. We compared foot outcomes in patients provided tele-consultations (April 2020-September 2020) with those who attended foot clinic (face-to-face) consultations in the pre-pandemic period (April 2019- September 2019).

## Research design and methods

### Participant inclusion

**During pandemic (Group 1).**   Patients regularly attending the multi-specialty diabetic foot clinic of a tertiary care hospital from north India for foot complications including (neuropathic foot with deformities, foot ulcer, ischemic foot, or Charcot neuroarthropathy prior to lockdown were the study cohort. The multi-disciplinary diabetic foot team comprised of an endocrinologist, infectious disease specialist, internist, general surgeon, radiologist, counsellor, and podiatrist. The demographic characteristics included age, gender, weight, height, body mass index (BMI), along with duration of diabetes, microvsacular and macrovascular complications, details of foot complications including the presence of neuropathy, peripheral arterial disease, presence of ulcer and ulcer characteristics and Charcot neuroarthropathy as well as laboratory parameters including HbA1c were prospectively entered in the electronic database during each visit prior to lockdown. Subsequently, the scheduled visits to the hospital were not possible due to lockdown, and patients were managed through telemedicine consultation for guidance and treatment. A verbal consent (video call and documented in case record form) was obtained from the patients in view of strict protocol of the management of COVID-19 (group 1 during April 2020- September 2020) and written informed consent was provided by the participants in group 2 for the use of medical records entered onto electronic repository during their physical visits to the hospital (April 2019 to September 2019). The study was approved by PGI Institute Ethics Committee reference no NK/2785/study/439 for the use of medical records of the patients included in the electronic repository and the telemedicine/virtual consultations (verbal consent) during ongoing COVID-19 pandemic. Virtual visit for podiatry care was non-existent in our hospital prior to COVID-19 pandemic.

### Protocol for video consultation

Patients with active foot ulcer after the declaration of lockdown were reviewed through video calls (Whatsapp, WeChat) for the assessment of glycemic status, foot complications (DFU etc) using pre-determined protocols that included clinical history, foot visual inspection (video-call) and education for domiciliary wound care. Patient were asked to lie supine in a well-lit room with light focussed on the foot (if possible) in the presence of the care giver (family member) and a local community nurse. Images and videos of the foot were acquired by the local team (community nurse and the care givers) and analysed in real time by the remote team (comprising of endocrinologist, internist, and surgeon). The community nurse was guided for cleaning the wound and obtain wound dimensions by transparent scale, wound dressing, obtaining swab (if clinically infected DFU), or assessment of temperature difference by dorsum of hand (suspected active Charcot) and to communicate offloading instructions to the patients in their local language. Because of the rapidly changing clinical situation, it was not possible to provide community health care providers with devices to measure foot temperature, so all were trained to assess differences in foot temperature using the dorsum of the hand and comparing affected foot with contralateral foot. Patients with prior foot complications but with no active foot ulcer at lockdown who sought tele-consultations were also

included in the study. A risk algorithm for the triage was followed as detailed elsewhere [18], so that the patients who were found to have large, infected ulcers, suspected osteomyelitis, diabetic foot infection (DFI) with systemic symptoms/signs of infection, gangrene or discoloration of foot were called for emergency hospital visit. DFI was clinically suspected based on classical signs and symptoms of infection in the absence of availability of tissue culture [19]. DFI in SARS-CoV-2 positive diabetic individuals or DFI with target organ damage like acute kidney injury or heart failure were counselled for early hospital visit.

**Exclusion criteria.** Patients requiring hospitalization for foot ailments or medical illness and with incomplete medical record in group 1 were excluded from the study.

**Before pandemic (Group 2).** Records of the patients with foot complications who obtained physical consultations in the same multi-specialty diabetic foot clinic of a tertiary care hospital from north India manned by a team comprising of an endocrinologist, infectious disease specialist, internist, general surgeon, radiologist, counsellor, and podiatrist during April 2019 to September 2019 were evaluated for the comparison of limb and life outcomes.

**Exclusion criteria.** Records of patients with life or limb-threatening foot infections, gangrene or with systemic complications requiring hospital admission was excluded from group 2.

## Definitions

The details of coexisting micro- and macrovascular complications of diabetes was retrieved from the electronic medical records (EMR) as documented at the last presentation before COVID-19 pandemic. Diabetic nephropathy was defined based on eGFR <60 ml/min/1.73m2 (calculated from CKD-EPI equation) and/or the presence of microalbuminura (24-hour urine albumin of 30–300 mg) or macroalbuminuria (urine albumin of >300 mg) and the details of retinopathy was obtained on detailed fundus examination. Peripheral arterial disease (PAD) and ischemic foot was defined as the absence of pedal pulses and/or Ankle-Brachial Index (ABI) <0.9 or prior history of revascularization of lower limb arteries; coronary artery disease (CAD) (history of coronary intervention/ revascularisation, pathological findings on coronary angiography, or prior records documenting CAD), cerebrovascular disease (CVD) and presence or absence of hypertension were also noted. Details of objective neurological examination that included Vibration Perception Threshold (VPT) and/or 10 gm-monofilament perception at 5 standardized plantar sites; and vascular examination with palpation of pedal pulses and ABI on both feet were noted. Diabetic peripheral neuropathy was considered as "present" in the presence of symptoms suggestive of peripheral neuropathy with VPT of more than 25 mV and/or absence of monofilament perception at any of the sites over feet tested.

## Outcome measure

The outcomes for ulcer and affected limb as well as mortality during the 24 weeks of lockdown from 25th March 2020 until 30th September 2020 were noted and compared to outcomes observed during a similar period of 2019.

## Statistical analysis

Data analysis was performed using the Statistical Package of Social Sciences (SPSS) version 23 (IBM Corp., Armonk, NY). Normality of the baseline variables was examined using the Shapiro-Wilk test and expressed as the median and interquartile range (IQR) for non-parametric data. The data of the entire cohort with foot complications under regular follow up prior to lockdown was further divided based on the presence or absence of active foot ulcer at the time of or during lockdown. The difference in ulcer outcomes amongst people with either

neuropathic or ischemic ulcer or ulcer with chronic Charcot foot is compared using Pearson chi-square test. P<0.05 was considered significant for the study.

## Results

Group 1 included 561 patients with prior foot complications following up at foot clinic sought teleconsultations (group 1) during the study period of COVID-19 pandemic. Twelve patients were found to have severe DFU (virtual triage) requiring hospital admissions were excluded from the analysis. Amongst 549 patients during the study period, median age was 57 (51 to 63) years and duration of diabetes was 10 (5 to 16) years. The foot complications included. 357 (65%) with neuropathic foot, 104 with predominant ischemic foot (18.9%) and 88 (16%) had chronic Charcot foot with deformities. 227 of 549 (41.3%) participants had active ulcer prior to lockdown and 32 patients without active ulcer in March 2020 (10.2%) developed a new-onset foot ulcer during lockdown (DFU group) and 290 did not have ulcer (non-DFU group). The demographic variables, diabetic complications, and other baseline characteristic (last available prior to lockdown visit) of the entire cohort are shown in Table 1. The baseline characteristics of participants with (n = 259) and without DFU (n = 290) at the start of the pandemic lockdown were similar (Table 2). In group 2, 650 patients presented with foot complications (21 requiring hospital admission were excluded) with median age 55(50–62) years and duration of diabetes 11 (5–15) years. 70.6% had neuropathic foot, ischemic foot in 16%, chronic Charcot foot in 13.4%. 366 patients had active foot ulcers [51 (14%) with new-onset ulcers] in group 2.

The median teleconsultations provided/required for patients in group 1 during the study period was 6 (2–11) in the DFU group and 1(1–3) in non-DFU group (p<0.01). The FBG was 124 mg/dl (101 to 167) and PPBG was 171 mg/dl (133 to 234.2) at last virtual consultation during the lockdown. In group 1, amongst patients with active foot ulcer, wound healing was observed in 93 patients (35.9%) and decrease in size in another 110 patients (42.5%), overall wound improvement in 78.4% as compared to wound healing in 32.8%, decrease in wound size in 43.2% patients, overall improvement of wound in 76% in group 2 (p = 0.318) as shown in Fig 1. Wounds were more likely to heal in those with neuropathic DFU as compared to either ischemic DFU or DFU associated with Charcot foot in both groups with no difference between groups (p = 0.07). In group1, 14 (5.4%) patients required amputations including 3 major amputations (below knee, all had ischemic ulcer) and 11 minor amputations (toe). Amputations were significantly more in patients with ischemic foot ulcers as compared to those with neuropathic DFU or Charcot foot with DFU (p<0.01).

Twenty-one patients (3.8%) died during the lockdown period in group 1 with six deaths in DFU group (5 had ischemic DFU and one had neuropathic DFU) related to DFI and acute coronary event as compared to 28 deaths (3.3%, p = 0.532) in group 2 (7 related to DFI, 14 due to acute coronary events and 7 due to renal cause). In group 1, 15 deaths occurred in non-DFU group that were unrelated to foot complications (ischemic cardiac events in 9, renal causes in 3, hyperosmolar coma in 2 and one participant with undefined reason).

## Discussion

We present outcomes of foot complications during the COVID-19 pandemic and subsequent lockdown in patients prospectively assessed by virtual triage and provided tele-consultations. We have shown that despite lack of face-to- face consultations, three-fourth of the patients with DFU experienced improvement in wound ulcer including closure or reduction in size of wound and only 5% of patients required amputations, predominantly minor amputations patients managed by remote consultations. There was no difference in foot outcomes observed

**Table 1. Demographic variables, diabetic complications, and other characteristics at baseline of the studied cohort.**

| Parameters | Group 1 | Group 2 | p-value | z- value |
|---|---|---|---|---|
| | n = 549 | n = 650 | | |
| Age(years) | 57 (51 to 63) | 55 (50 to 62) | 0.23 | -2.278 |
| HbA1c (mmol/mol) | 8.5(7.2 to 10.3) | 8.4 (7.0 to 10.2) | 0.355 | -0.925 |
| Egfr (ml/min/1.73 m$^2$) | 72.9 (50.6 to 92.9) | 66.8 (41.4 to 89.1) | 0.005* | -2.821 |
| Duration (years) | 10 (5 to 16) | 11.0 (5.0 to 15.0) | 0.098 | -1.653 |
| BMI (kg/m$^2$) | 24.3 (21.8 to 28) | 24.2 (21.5 to 27.7) | 0.568 | -0.572 |
| FBG (mg/dL) | 133 (105 to 180) | 143.3 (104.0 to 196.0) | 0.257 | -1.133 |
| PPBG (mg/dL) | 194 (150 to 259.7) | 203.5 (153.2 to 269.7) | 0.095 | -1.672 |
| 24 hours urine protein(mg/dL) | 289 (120 to 757) | 390.6 (169.2 to 1000) | 0.003 | -2.929 |
| Cholesterol (mmol/L) | 135 (110 to 170) | 142 (116.0 to 172.3) | 0.195 | -1.297 |
| Hb (g/L) | 10.7(9.2 to 12.1) | 10.0 (8.6 to 11.5) | 0.000* | -4.775 |
| LDL(mmol/L) | 74 (49 to 98.5) | 79.4 (57.7 to 103.0) | 0.015* | -2.440 |
| HDL(mmol/dL) | 40 (33 to 47) | 39.0 (30.0 to 45.0) | 0.009 | -2.623 |
| TG (mmol/L) | 120 (96 to 156) | 127 (96.0 to 174.1) | 0.110* | -1.600 |
| Nephropathy, n (%) | 154 (28.05) | 253(38.92) | 0.000* | - |
| Retinopathy, n (%) | 263 (47.90) | 391 (60.15) | 0.000* | - |
| Neuropathy, n (%) | 510 (92.28) | 590 (90.76) | 0.182 | - |
| CAD, n (%) | 115 (20.94) | 108 (16.61) | 0.547 | - |
| Stroke, n (%) | 37 (6.73) | 36 (5.53) | 0.386 | - |
| HTN, n (%) | 254 (46.26) | 384 (59.07) | 0.000* | - |

Data represented as median (inter-quartile range) unless specified and two group comparison was performed by Mann-Whitney test. Categorical variable are represented as n (%) and intergroup comparison was performed by Chi-square test.

*p<0.05 was considered significant

Egfr- estimated glomerular filtration rate, BMI–Body mass index, FBG- Fasting blood glucose, PPBG- Post prandial blood glucose, Hb–Haemoglobin, LDL- Low density lipoprotein, HDL–High density lipoprotein, TG–Triglycerides, CAD- Cardiovascular disease, HTN–Hypertension, z- Standard score

in people with diabetic foot complications who were evaluated face-to face and treated in the foot clinic previous year and those managed through virtual means during the pandemic, suggesting efficacy of video consultations for managing diabetic foot complications.

It is known that people with diabetes and foot ulcers have worse outcomes including amputations and mortality as compared to those without foot ulcers. Amongst patients with DFU, those with ischemic DFU have poorer outcomes with higher amputations and mortality rates compared to neuropathic DFU, as was also observed in the present study. Thus, people with active or healed DFU require regular follow up for foot examination, evaluation of risk factors to ensure remission of DFU. The COVID-19 pandemic has suddenly interrupted patient foot education, diagnosis and treatment of foot complications due to the suspension of out-patient clinics and hospital visits. Moreover, patients with diabetes also face difficulties in procuring medicines including insulin as well as dressings because of interrupted supplies in the COVID-19 pandemic [11]. The role of self-care cannot be over-emphasized in such situations and tele-consultation is one of the options that has been shown to assist physicians to provide proper guidance and remote examination of the patients with diabetic foot complications [20]. Thus, we contemplated the present study to understand the ulcer and limb outcomes for people with active or healed DFU who were provided video tele- consultations during the COVID-19 pandemic.

In our study cohort only 5.4% patients with mild to moderate DFU required amputation (predominantly minor amputations) during the COVID-19 pandemic. A hospital based study

**Table 2. Baseline characteristic of studied cohort of group 1 stratified by the presence or absence of active DFU.**

| Parameters | Active DFU Group | Non- DFU group | p-value | z-value |
|---|---|---|---|---|
| | n-259 | n- 290 | | |
| Age (years) | 57 (52 to 63) | 57 (51 to 63) | 0.439 | -0.779 |
| HbA1c (%) | 8.3 (7.2 to 10) | 8.8 (7.3 to 10.6) | 0.021* | -2.315 |
| eGFR (ml/min/1.73m$^2$) | 73.6 (53.4 to 95.4) | 72.5 (45.7 to 90.4) | 0.193 | -1.301 |
| Duration (years) | 10 (4.5 to 15) | 11 (5 to 17) | 0.309 | -1.018 |
| BMI (kg/m$^2$) | 24.2 (21.9 to 28) | 24.4 (21.1 to 28.1) | 0.676 | -0.418 |
| FBS (mg%) | 132 (100.5 to 181) | 133 (108 to 180) | 0.303 | -1.030 |
| PP (mg%) | 193 (150 to 252) | 198 (149.5 to 264.3) | 0.449 | -0.756 |
| Urine protein (mg/ 24 hour) | 250 (110 to 735) | 300 (150 to 820) | 0.216 | -1.239 |
| Cholesterol (mg%) | 140 (112.0 to 176.5) | 134 (107.6 to 160.8) | 0.000* | -17.325 |
| Hb (g/L) | 11.1 (9.3 to 12.4) | 10.4 (9.1 to 11.8) | 0.000* | -17.428 |
| LDL (mg%) | 75.5 (53.1 to 99.7) | 72 (46 to 97) | 0.166 | -1.386 |
| HDL (mg%) | 40.7 (33.2 to 48.1) | 39.5 (32 to 46) | 0.137 | -1.487 |
| TG (mg%) | 122 (96.0 to 153) | 120 (94 to 160) | 0.947 | -0.67 |
| Nephropathy, n (%) | 76 (29.3) | 78 (26.8) | 0.582 | - |
| Retinopathy, n (%) | 107 (41.3) | 156 (53.7) | 0.003 | - |
| Neuropathy, n (%) | 240 (92.6) | 270 (93.1) | 0.874 | - |
| CAD, n (%) | 51 (19.6) | 64 (22.0) | 0.349 | - |
| Stroke, n (%) | 19 (7.3) | 18 (6.2) | 0.691 | - |
| HTN, n (%) | 126 (48.6) | 128 (44.1) | 0.334 | - |

Data represented as median (inter-quartile range) unless specified and two group comparison was performed by Mann-Whitney test. Categorical variable are represented as n (%) and intergroup comparison was performed by Chi-square test.

*p<0.05 was considered significant

eGFR- estimated Glomerular filtration rate, BMI -Body mass index, FBS- Fasting blood glucose, PPBG- Post prandial, Hb—Haemoglobin, LDL- Low density lipoprotein, HDL- High density lipoprotein, TG—Triglycerides, CAD—Cardiovascular disease, HTN–Hypertension, z- Standard score.

of patients coming to emergency department with severe DFU during COVID-19 pandemic observed three-fold higher risk of limb amputation as compared to the similar period in year 2019 (60% vs 18%, p = 0.001) [21]. Another study observed a similar number (24.25%) of minor amputation following foot ulcers during COVID-19 pandemic compared to (20.5%, p>0.05) pre-pandemic period [22]. Other authors have also noticed an increased rates of

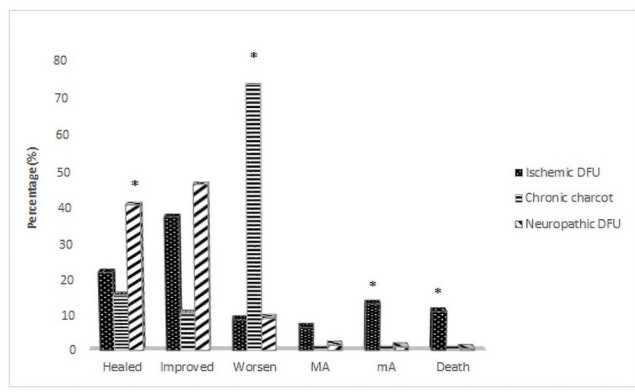

**Fig 1. Ulcer and limb outcomes stratified by type of diabetic foot complications during COVID-19 pandemic.**
MA: Major amputation; mA: Minor Amputation; * p<0.05 and significant.

major or minor amputations during the pandemic period attributed to predominantly ischemic ulcers or non infected DFU [23–25]. The amputation rates in our study was significantly less as compared to that observed by Caruso P, et al [21] and Schmidt et al [22] despite the inclusion of patients with active DFU (either ischemic or neuropathic) with or without DFI. Probable reasons for lesser amputation rates may be because we excluded patients (twelve in total) with limb threatening DFU, and those with suspected osteomyelitis as well as those with sign or symptoms of necrotizing fasciitis, SIRS or gangrene on video examination. Secondly, the included patients were under regular follow up in our foot clinic prior to lockdown, had foot care awareness and were motivated for self-examination of foot. More than half of the patients (58%) in the present cohort did not have active DFU but had prior DFU and were at high risk of recurrence. We observed new onset DFU in only 10.2% of these patients with prior DFU which is comparable to the previous year (14%) and no amputations during the 24-week study observation period.

Recently, integration of virtual care and EMR technologies (STRIDE protocol) has been shown to help in rapid triage, effective delivery of foot care, and limb salvage [22]. Another internet-based algorithm and online consultation service by instant messaging has been proposed as a feasible option for grading of the wound, risk categorization and modulation of home-care [26]. A further modality for the effective management of patients with active DFU during pandemic time is through fast-track pathway (FTP) classification [15]. The FTP classification is considered an easy tool for non-specialists health care providers in primary care settings for global assessment of DFU. A study of patients with active DFU from Italy observed that majority (54.7%) of the ulcers reporting to hospital during COVID-19 pandemic are either neuroischemic or ischemic and 42% also involved gangrene [27]. Utilizing FTP pathway for patient triage, Meloni et al observed healing of DFU in 27.1% and amputations in 3.7% of patients with active DFU during the median follow up of 42 days of COVID-19 pandemic [15]. The limb outcomes using FTP classification are like the present study, barring that the setting for delivery of foot care is different as we provided virtual foot care delivery compared to hospital-based triage and management of foot complications. The low amputation rate in the present study assures about the efficacy of remote examination, video consultations and diligent counselling in improving limb outcomes for people with active or healed DFU. In addition, importance of glycemic control need to be reinforced in people with diabetes and foot complications [28].

This is the first longitudinal study depicting limb salvage in patients with prior foot complications provided virtual foot care consultations and comparison with the cohort who had face-to-face foot clinic consultations in the previous year (pre-pandemic period). The risk stratification and wound assessment was performed virtually by a single investigator experienced in wound management and guidance was provided after deliberation with the multi-disciplinary team. During pandemic, wound care was performed by locally available trained personnel and caregivers unlike specialized centre wound management in the pre-pandemic year. The off-loading modalities were not similar across the group and compliance was at best ensured through repeated video consultations and total contact cast could not be provided for those with DFU. Hence, some of the active DFU may have been categorized as non-infected DFU, as tissue culture was not possible in the patients included in the present study because of the lack of expertise at the local levels or in domiciliary care settings. Many diabetic foot wounds may lack sign of infection on visual inspection and requires additional investigations including serological and radiological tests. Other limitations inherent to the study including lack of validated remote monitoring tool or automated software for wound measurement to objectively assess the rate of change in wound area during domiciliary care setting. Hence, the result of the present study is applicable to select subset of patients who are under regular follow up and

seek virtual podiatry consultations rather than to new patients of foot complications. The treatment modalities in the two groups were different in view of domiciliary care (group 1) compared to foot clinic care (group 2). The outcomes were studied over a short span of 20 weeks and may not be enough to assess the full impact of the pandemic and restricted foot care on limb outcomes and to determine the outcome differences between groups.

In conclusion, targeted foot care service delivery through video consultations is feasible, may help in reducing the need for hospital visit, and may achieve similar limb and life outcomes in motivated patients with foot complications. Though foot care services remain gravely affected during COVID-19 pandemic; however, people with uncomplicated DFU and at-risk for DFU including those with healed ulcers requiring uninterrupted clinic care can be judiciously managed through personalized virtual communications.

## Supporting information

**S1 File.**
(XLSX)

**S2 File.**
(XLSX)

## Acknowledgments

We acknowledge Miss Raveena, Mrs Reshma for assisting in data collection.

## Author Contributions

**Conceptualization:** Ashu Rastogi, Anil Bhansali A., Edward B. Jude.

**Data curation:** Ashu Rastogi, Priya Hiteshi.

**Formal analysis:** Ashu Rastogi, Priya Hiteshi.

**Investigation:** Ashu Rastogi, Priya Hiteshi.

**Methodology:** Ashu Rastogi.

**Project administration:** Ashu Rastogi.

**Supervision:** Anil Bhansali A., Edward B. Jude.

**Validation:** Ashu Rastogi.

**Writing – original draft:** Ashu Rastogi, Priya Hiteshi.

**Writing – review & editing:** Ashu Rastogi, Anil Bhansali A., Edward B. Jude.

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
