## [Decision Letter · Decision Letter 0]

8 Apr 2021

PONE-D-21-06327

Virtual Triage and Outcomes of Diabetic Foot Complications during COVID-19 Pandemic: A Retro-Prospective, Observational cohort study

PLOS ONE

Dear Dr. Rastogi,

Thank you for submitting your manuscript to PLOS ONE. After careful consideration, we feel that it has merit but does not fully meet PLOS ONE’s publication criteria as it currently stands. Therefore, we invite you to submit a revised version of the manuscript that addresses the points raised during the review process.

We look forward to receiving your revised manuscript.

Kind regards,

Kanhaiya Singh, Ph.D

Academic Editor

PLOS ONE

Journal Requirements:

3. We note that you obtained verbal consent from Group 1. In the Methods, please clarify whether consent was given to participate in this study or for another reason. Please also state in the Methods:

- Whether the Institutional Review Board (IRB) approved use of oral consent

- How oral consent was documented

For more information, please see our guidelines for human subjects research: https://journals.plos.org/plosone/s/submission-guidelines#loc-human-subjects-research

4. In addition, in your ethics statement in the manuscript please ensure that you have discussed whether all data/samples from Group 2 were fully anonymized before you accessed them and/or whether the IRB or ethics committee waived the requirement for informed consent. If patients provided informed written consent to have data/samples from their medical records used in research, please include this information.

5. In your Methods section, please provide additional information about the participant recruitment method and the demographic details of your participants. Please ensure you have provided sufficient details to replicate the analyses such as descriptions of where participants were recruited and where the research took place.

6. Please include the full name of the ethics committee that reviewed and approved your study in the manuscript Methods.

7. We note that you have indicated that data from this study are available upon request. PLOS only allows data to be available upon request if there are legal or ethical restrictions on sharing data publicly. For information on unacceptable data access restrictions, please see http://journals.plos.org/plosone/s/data-availability#loc-unacceptable-data-access-restrictions.

Additional Editor Comments:

Please expand the table and figure legends including details like how data was represented (e.g. mean +/- standard deviation). Please provide values of statistical tests in addition to p value (e.g. odds ratio or other values). What was the power of study? Please also provide details about the human subject research approval of the study in the method part including the approval number and/or a statement indicating approval of this research. Also, the exclusion criteria of the patients should be presented as a separate section.

Reviewers' comments:

Reviewer's Responses to Questions

**Comments to the Author**

1. Is the manuscript technically sound, and do the data support the conclusions?

Reviewer #1: Yes

Reviewer #2: Yes

2. Has the statistical analysis been performed appropriately and rigorously? 

Reviewer #1: Yes

Reviewer #2: Yes

3. Have the authors made all data underlying the findings in their manuscript fully available?

Reviewer #1: Yes

Reviewer #2: Yes

4. Is the manuscript presented in an intelligible fashion and written in standard English?

Reviewer #1: Yes

Reviewer #2: Yes

5. Review Comments to the Author

Reviewer #1: The authors have submitted an interesting study wherein they have shown that teleconsultation with video call for visual inspection of the ulcers is very helpful in the care of Diabetic patients with foot ulcers or those who are at risk during the lockdown period necessitated by the Corona pandemic. They have shown that outcomes and wound healing rates were similar to a cohort of patients who had physical follow-ups in the same period a year earlier i,e. pre-lockdown. The authors have also discussed the limitations of this study because it would be applicable in a select subset of patients.

Reviewer #2: The Manuscript “Virtual Triage and Outcomes of Diabetic Foot Complications during COVID-19 Pandemic: A Retro-Prospective, Observational cohort study” by Dr. Rastogi et al, Compares the care provided to patients with diabetic foot by virtual consultation during the COVID 19 pandemic and face-to-face care provided to the diabetic patients. It mainly focuses on the outcomes of diabetic foot and developing complications like foot ulcers in patients provided tele-consultations with those who attended foot clinic consultations in the pre-pandemic period. This research tried to elucidate the benefits of remote consultations over in-clinic consultations.

Allocation of patients in each group is appropriate for the study. Size of both groups is large enough for efficient statistical comparison of the treatment. Patient follow-up and wound assessment were performed by a single investigator, and trained personnel, would decrease the subjective error in the study. Because of the limitation due to COVID-19, investigations like difference in foot temperature and other DFU estimations are done by trained personnel at patients’ resident. Visual estimation of wound infection and wound healing, without appropriate investigations may limit the study but, author tried to address these limitations and overcome it by using the appropriate methods. Though the study duration is less, long enough to address the significance of the study. It would be more appropriate for author to include the parameters like HbA1c, FBG, PPBG and other complications like HTN and nephropathy at the end of the study. These parameters will reflect the patient’s compliance to diabetic treatment and give us the actual outcome of the foot ulcers.

This is manuscript is technically sound enough and the statistically significant data supports it.

I recommend that this paper be accepted.

6. PLOS authors have the option to publish the peer review history of their article (what does this mean?). If published, this will include your full peer review and any attached files.

Reviewer #1: **Yes: **Sanjeev Kumar Gupta

Reviewer #2: No

---

## [Author Response · Author response to Decision Letter 0]

16 Apr 2021

Sir,

 I would like to thank you for your time during the ongoing COVID-19 pandemic giving an opportunity to revise the manuscript. I sincerely thank the editor and reviewers for their suggestions and constructive comments. We have revised the article as per your suggestions and provided the changes highlighted BLUE in the revised manuscript.

Editor Comments:

Please expand the table and figure legends including details like how data was represented (e.g. mean +/- standard deviation). Please provide values of statistical tests in addition to p value (e.g. odds ratio or other values). What was the power of study? Please also provide details about the human subject research approval of the study in the method part including the approval number and/or a statement indicating approval of this research. Also, the exclusion criteria of the patients should be presented as a separate section.

Reply: The table and figure legends are expanded as detailed. Odds ratio were not calculated for independent group comparisons.

This was an observational retro-prospective study during the present ongoing COVID19 pandemic. Power of the study was not obtained.

Statement about research approval is provided in the revised manuscript (Method section)

Exclusion criteria are provided in separate section as suggested (Method section)

Reviewers' comments:

Reviewer's Responses to Questions

Comments to the Author

1. Is the manuscript technically sound, and do the data support the conclusions?

Reviewer #1: Yes

Reviewer #2: Yes

Reply: We thank you for your comments

2. Has the statistical analysis been performed appropriately and rigorously?

Reviewer #1: Yes

Reviewer #2: Yes 

Reply: We thank you for your comments

3. Have the authors made all data underlying the findings in their manuscript fully available?

Reviewer #1: Yes

Reviewer #2: Yes

Reply: We thank you for your comments

4. Is the manuscript presented in an intelligible fashion and written in standard English?

Reviewer #1: Yes

Reviewer #2: Yes

Reply: We thank you for your comments

5. Review Comments to the Author

Reviewer #1: The authors have submitted an interesting study wherein they have shown that teleconsultation with video call for visual inspection of the ulcers is very helpful in the care of Diabetic patients with foot ulcers or those who are at risk during the lockdown period necessitated by the Corona pandemic. They have shown that outcomes and wound healing rates were similar to a cohort of patients who had physical follow-ups in the same period a year earlier i,e. pre-lockdown. The authors have also discussed the limitations of this study because it would be applicable in a select subset of patients.

Reply: We thank you for your comments

Reviewer #2: The Manuscript “Virtual Triage and Outcomes of Diabetic Foot Complications during COVID-19 Pandemic: A Retro-Prospective, Observational cohort study” by Dr. Rastogi et al, Compares the care provided to patients with diabetic foot by virtual consultation during the COVID 19 pandemic and face-to-face care provided to the diabetic patients. It mainly focuses on the outcomes of diabetic foot and developing complications like foot ulcers in patients provided tele-consultations with those who attended foot clinic consultations in the pre-pandemic period. This research tried to elucidate the benefits of remote consultations over in-clinic consultations.

Allocation of patients in each group is appropriate for the study. Size of both groups is large enough for efficient statistical comparison of the treatment. Patient follow-up and wound assessment were performed by a single investigator, and trained personnel, would decrease the subjective error in the study. Because of the limitation due to COVID-19, investigations like difference in foot temperature and other DFU estimations are done by trained personnel at patients’ resident. Visual estimation of wound infection and wound healing, without appropriate investigations may limit the study but, author tried to address these limitations and overcome it by using the appropriate methods. Though the study duration is less, long enough to address the significance of the study. It would be more appropriate for author to include the parameters like HbA1c, FBG, PPBG and other complications like HTN and nephropathy at the end of the study. These parameters will reflect the patient’s compliance to diabetic treatment and give us the actual outcome of the foot ulcers.

This is manuscript is technically sound enough and the statistically significant data supports it.

I recommend that this paper be accepted.

Reply: We thank you for your comments. 

The detailed evaluation regarding nephropathy and hypertension was not possible during lockdown for patients in group 1 were provided virtual teleconsultations. 

FBG and PPBG values were provided on home based self-monitoring of blood glucose in the revised manuscript. 

HbA1c was not possible due to same reasons for lack of clinic visit.

We hope that the revised manuscript may be considered for publication.

Sincerely,

Ashu Rastogi

---

## [Editor Report · Decision Letter 1]

21 Apr 2021

Virtual Triage and Outcomes of Diabetic Foot Complications during COVID-19 Pandemic: A Retro-Prospective, Observational cohort study

PONE-D-21-06327R1

Dear Dr. Rastogi,

We’re pleased to inform you that your manuscript has been judged scientifically suitable for publication and will be formally accepted for publication once it meets all outstanding technical requirements.

Kind regards,

Kanhaiya Singh, Ph.D

Academic Editor

PLOS ONE
---

## [Editor Report · Acceptance letter]

26 Apr 2021

PONE-D-21-06327R1 

Virtual triage and outcomes of diabetic foot complications during Covid-19 pandemic: A retro-prospective, observational cohort study 

Dear Dr. Rastogi:

I'm pleased to inform you that your manuscript has been deemed suitable for publication in PLOS ONE. Congratulations! Your manuscript is now with our production department. 

Kind regards, 

on behalf of

Dr. Kanhaiya Singh 

Academic Editor

PLOS ONE